# Applications and Prospects of Nanotechnology in Food and Cosmetics Preservation

**DOI:** 10.3390/nano12071196

**Published:** 2022-04-03

**Authors:** Paraskevi Angelopoulou, Efstathios Giaouris, Konstantinos Gardikis

**Affiliations:** 1IPSP Nanomedicine, Medical & Pharmacy Department, School of Health Sciences, National and Kapodistrian University of Athens, 15772 Athens, Greece; paraskevi.agg@gmail.com; 2Laboratory of Food Microbiology and Hygiene, Department of Food Science and Nutrition, School of the Environment, University of the Aegean, 81400 Myrina, Greece; stagiaouris@aegean.gr; 3R&D Department, APIVITA SA, Industrial Park, Markopoulo, 19003 Athens, Greece

**Keywords:** food preservation, cosmetics preservation, nanotechnology, active packaging

## Abstract

Cosmetic and food products containing water are prone to contamination during the production, storage, and transit process, leading to product spoilage and degraded organoleptic characteristics. The efficient preservation of food and cosmetics is one of the most important issues the industry is facing today. The use of nanotechnology in food and cosmetics for preservation purposes offers the possibility to boost the activity of antimicrobial agents and/or promote their safer distribution into the end product upon incorporation into packaging or film constructions. In this review, current preservation strategies are discussed and the most recent studies in nanostructures used for preservation purposes are categorized and analyzed in a way that hopefully provides the most promising strategies for both the improvement of product safety and shelf-life extension. Packaging materials are also included since the container plays a major role in the preservation of such products. It is conclusively revealed that most of the applications refer to the nanocomposites as part of the packaging, mainly due to the various possibilities that nanoscience offers to this field. Apart from that, the route of exposure being either skin or the gastrointestinal system involves safety concerns, and since migration of nanoparticles (NPs) from their container can be measured, concerns can be minimized. Conclusion: Nanomaterial science has already made a significant contribution to food and cosmetics preservation, and rapid developments in the last years reinforce the belief that in the future much of the preservation strategies to be pursued by the two industries will be based on NPs and their nanocomposites.

## 1. Introduction

Like any other product containing water and organic or inorganic substances, cosmetic and food products may be contaminated during the production, storage, and transit process, leading to product spoilage and degraded organoleptic characteristics, and probably causing adverse effects to the consumer. Preservation is a way of processing goods in order to avoid their spoilage and extend their shelf life, and it is mainly achieved in three ways: (i) protecting from microbial contamination and growth, (ii) protecting from oxidation, and (iii) preventing enzymatic or non-enzymatic browning. Ancient people first introduced the importance of preservation while storing their food in cold and dark places. Preservation can be direct to the bulk product—for example, with the addition of a preservative—and/or indirect by the use of the appropriate packaging.

The selection of the appropriate preservative depends on several factors. First of all, its actual use and the route of exposure direct the choice. The ideal preservative should be foremost non-toxic and safe for the consumer. Cosmetic products intended for use around the eyes, baby products, or intimate care products have usually different limitations regarding the preservatives they incorporate. The restriction of some preservatives is due to the fact that the extent, duration, frequency, and route of exposure can have a significant impact on the toxicity of a compound. Even though preservatives are added in a cosmetic formula to ensure their safety to the consumer, some reactions of tolerance during usage are observed from most of them in the current market. These reactions range from mild irritation of the skin to estrogenic activity, which in some cases has been correlated with mammary tumors [1]. On the other hand, there is no need to add a bacteriostatic preservative in a food that does not support bacterial growth due to its endogenous properties (e.g., low pH); however, the same food may require a fungi-oriented preservative (e.g., sorbic acid and its salts) [2].

Furthermore, the preservative or preservative system should be stable, compatible, cost effective (preferably in low concentrations), and should not alter the organoleptic properties of the final product. Several factors can affect the stability of a preservative. High solubility and a good O/W partition coefficient will enhance its activity in the aqueous phase of the formula [3]. The pH of the formula can provoke the decomposition of the antimicrobial agent or modify its conservating activity [4]. The temperature during the manufacturing process or during product use can affect preservative stability in a much higher degree when the compound is volatile. 

A preservative must be highly compatible with other ingredients of the formula, such as surfactants, solvents, perfumes, or active compounds and other food additives. Some preservatives can be inactivated because of the antagonistic action of another compound. The presence of non-ionic surfactants can alter the antimicrobial activity of parabens. Solid materials or organic solids such as cellulose and starch can absorb preservatives when they are in high concentrations. On the other hand, some polyols and sunscreen filters can boost preservative efficacy. Chelating agents such as ethylenediaminetetraacetic acid (EDTA) are well known to act as preservative boosters. Physical compatibility is also important. The preservative must be colorless and odorless and not alter the organoleptic characteristics of the final formula.

Several microorganisms are equipped with mechanisms of resistance to the preservatives. The inactivation of the preservative agent, the reduction in preservative efficacy, or a tolerance of microorganisms is called preservative resistance [5]. Scientists claim that the reason why this is happening is the use of preservatives in very small quantities due to their high toxicity and thus the progressive adaptation of microbial cells to them [6]. It has been observed that the most resistant forms are bacterial endospores [7]. Mycobacteria are more resistant than Gram-negative bacteria, and it is easier for preservatives to affect Gram-positive bacteria [7]. Preservative resistance increases the urgency for the development of new antimicrobial agents or preservation techniques to overcome this phenomenon.

Toxicity of nanomaterials (NMs) is a hot topic that influences and affects both research studies and regulatory status. NPs can enter the human body most often through the respiratory or gastrointestinal tract, skin, or through systemic administration, depending on the route of exposure [8].

Several studies have proven the accumulation of NPs on different human organs, namely, the liver, lungs, spleen, and others. On a cellular level, when in contact with cells, NPs are normally ingested by endocytosis and are captured inside endosomes or lysosomes without entering the cytoplasm [9].

In order to take chemical information and to support in vitro techniques that are used to study the distribution of NPs into cells, SR-X-ray absorption near edge structure (XANES) can be applied. For example, this technique showed that silver NPs were gradually transformed into ionic silver inside cultured macrophages [10].

In vitro studies have been conducted to show the effect of NP accumulation on the cell cytoskeleton. The organization of cytoskeletal actin or F-actin orientation after the exposure of human fibroblasts on gold NPs have been defined by flow cytometry [11].

At this point, it is important to emphasize that the greatest destructive effect of NPs when entering the cell is free radical production, which is found to be increased to 350% when exposed to silver NPs for 48 h. More specifically, that effect causes mitochondrial membrane disruption and cell viability reduction as a consequence. In a recent study, a depolarization phenomenon on mitochondrial membrane due to a silver NP reaction was visualized by flow cytometry, and for cellular viability reduction scientists applied the viability assay WST-8 [12].

## 2. Applications of Nanotechnology in the Cosmetics and Food Industries

In general, the applications of nanotechnology for the preservation of cosmetics and food concern NPs, nano-delivery systems, and nanocomposites. Properties, the mode of action, and the most promising studies are presented. Since packaging plays a crucial role in the preservation of both cosmetics and food, and taking into consideration that over 95% of the literature available till now reports applications of NMs in packaging for preservation purposes, the following section also analyzes the most recent advances in this field in depth.

### 2.1. Food and Cosmetics Packaging

According to the Commission Implementing Decision 2013/674/EU, «packaging material is the container (or primary packaging) that is in direct contact with the formulation» [13]. Primary packaging plays an important role in the preservation and safety of the cosmetic/food product not only because it protects it from microbial contamination and subsequent spoilage, but because at the same time it may also interact with the product, either through the migration of substances it may contain (including antimicrobials), or through the transport of atmospheric agents such as oxygen in the product. This is the reason why the cosmetic file should contain specific characteristics of the primary container such as composition, possible impurities, and possible migration. Moreover, compatibility tests with the cosmetic product and composition are mandatory. The EU Framework Regulation for food contact materials (Regulation (EC) No 1935/2004) requires that materials be manufactured according to Good Manufacturing Practices (GMP) and not release their constituents into food at levels harmful to human health, and provides rules for compliance documentation and traceability [14]. The European Union essentially relies on the relevant legislation on food packaging and acknowledges that once a packaging material has been accepted for food, then it is more easily approved for cosmetics as well [13].

#### 2.1.1. Smart Packaging

There are four ways to improve food packaging, making it “smart”: (i) the improvement of its mechanical and barrier properties; (ii) the delivery of antimicrobials that slowly release into the product; (iii) the incorporation of sensors that can detect harmful substances, microbial spoilage, or gas; and finally, (iv) the development of a packaging made from biopolymers. 

##### Active and Intelligent Packaging

Active packaging technology is used to extend products’ shelf life by incorporating preservatives, oxygen scavengers, moisture absorbers, carbon dioxide emitters, and ethylene scavengers into the packaging material. Active packaging creates a microenvironment between food and packaging materials that can scavenge oxygen or moisture and prevent the evaporation of volatile substances such as flavors and ethanol, and it may also offer antimicrobial activity [15]. During the last few decades, a variety of antimicrobial agents have been incorporated into packaging materials, films, and coatings to extend the shelf life of packaged products and avoid microbial spoilage. This is specifically called “antimicrobial active” packaging and concerns packaging systems that deliver antimicrobial agents and release them within the product, in some cases in a controlled manner. Among biodegradable polymers, those that are used most to deliver preservatives in food are polylactic acid (PLA), cellulose, carrageenan, starch, and chitosan. In some cases, two or more different polymers are applied as a mixture to take advantage of their different properties in the end product [16,17].

A plethora of antimicrobial agents have been introduced in antimicrobial active packaging systems: nisin, pediocin, sodium benzoate, potassium sorbate, propyl paraben, antimicrobial essential oils, and other plant extracts [18,19,20,21,22], and their applications are found in different types of products, including fresh meat, fish, nuts, fresh fruits, beverages, and others. Recently, the cosmetics industry obtained active packaging technology from the food industry to prevent microbial spoilage [23].

Along with the active packaging, “intelligent” packaging has also been developed, and together they compose the term “smart” packaging. Intelligent packaging is a container, coating, or film that can detect impurities of dangerous substances, as well as biochemical or microbial changes in the product. This is achieved using sensors in the packaging, and the science behind this achievement is now based in nanotechnology. Two types of nano-sensors can be used in food packaging: electrochemical and optical. The incorporation of NMs into sensing systems imparts properties such as optical, thermal, plasmonic, catalytic, and others, improving their performance. Copper NPs have been described in the packaging of soft drinks and detect carbohydrate oxidation [24], graphene nanoribbons find applications in detecting various antioxidants in mixed fruit juices [25], carbon NPs detect the presence of melamine in milk [26], and gold NPs can detect heavy metals in water [27], and when they are electrodeposited on graphene ribbons, they can detect the presence of tert-butylhydroquinine in edible oils [28].

In the paragraphs hereunder, the different types of NMs that find applications in the cosmetics and food industries are presented. 

### 2.2. Nanocomposites Used in the Cosmetics and Food Industries

As already mentioned, the packaging of food and cosmetics plays an important role in their preservation by preventing the entry of germs and delaying their spoilage. The evolution of nanotechnology and the multitude of properties that NPs have, have further advanced the packaging technology with the creation of active and intelligent packaging. The release of preservatives from the container to the product has made it possible to extend the shelf life of the latter, as well as to reduce the concentration of chemical preservative systems within the product, thus improving its safety. Consumer demands for and positive environmental impact of biodegradable packaging and sustainable waste disposal management have led to the development of packaging, including films and coatings, using only biodegradable materials, also called biopolymers. Among these biopolymers, cellulose, carrageenan, agar, starch, and chitosan have been investigated most. However, extended research concluded that packaging production using biopolymers has drawbacks, mainly in terms of strength and permeability, known as mechanical and barrier properties. Systems that have been shown to improve these properties are nanocomposites.

A nanocomposite is a heterogeneous system of two or more constituents with different characteristics, chemical and physical, and one of the components has at least one dimension in nanoscale. Nanocomposites in packaging are based on biopolymers as the continuous phase and nanofillers as the discontinuous phase. When fabricated in nanoscale, biopolymers have special properties that make them ideal materials for creating antimicrobial active films, such as the so-called “bacterial” nanocellulose. This refers to a natural nontoxic biopolymer synthesized from special bacterial species with unique nanostructured morphology, providing elastic properties in combination with high surface area, crystallinity, porosity, and resistance. Within its pores, bacterial nanocellulose can be the guest of several molecules held from its hydroxyl groups, such as antibacterial agents. The second most abundant polysaccharide resource in nature is chitin, which is derived from shellfish waste, and it differs from cellulose in that it has an amino group at the C-2 position instead of a hydroxyl group. Its partially deacetylated form is called chitosan, and it can be functionalized in derivatives that contain cationic or other moieties. Among structural properties, chitosan and its derivatives present antimicrobial activity that is dependent on its molecular weight, cationic charge density and position, degree of acetylation, hydrophobicity, and other functional groups it may contain [29]. Different modes of action are observed between Gram-positive and Gram-negative bacteria due to the differences in their cell envelope composition. In both cases, antimicrobial activity is based on the cationic charges in the chitosan backbone. 

Nanofillers that have been incorporated into such composites include NPs, nanorods, nanotubes, and nanofibrils. The applications of different nanofillers for preservation purposes are summarized in Table 1.

### 2.3. Carbon-Based NMs—Fullerenes

Carbon-based NPs include graphene oxide, carbon dots, carbon nanotubes, and fullerenes, and they have already found applications in tissue engineering, drug delivery, imaging, diagnosis, and cancer therapy. In the cosmetics industry, carbon nanotubes are used as delivery systems for bioactive compounds. Fullerenes are valuable skin-rejuvenating agents due to their ability to scavenge free radicals about 172 times more than vitamin C [36]. Indeed, scientists are investigating their cytoprotective therapeutic potential for several dermatological recovery treatments [37]. Fullerenes are usually dispersed into squalene or wrapped in polyvinylpyrrolidone. Yet, the applications of both type of ingredients are limited. Regarding fullerenes, the main obstacle is their poor solubility; however, this can be overcome either by using their derivatives or by encapsulating them into liposomes [38]. The most commonly used fullerenes in the cosmetics industry are fullerene C60 and fullerene C70, and although they are approved as antimicrobial and skin-conditioning agents, they are also applied in anti-ageing, whitening, and sun care products [39,40,41]. Very recently, on 6 July 2021, the European Commission requested a scientific opinion on fullerenes and their derivatives from the Scientific Committee for Consumer Safety. This was triggered from 19 requested notifications for new cosmetic products containing fullerenes. The deadline for the results was set at six months [42].

Carbon dots, nanotubes, and graphene oxide are already finding applications that improve antimicrobial activities of active packaging. Regarding fullerenes, such application has not been reported yet. However, a promising study is examining the antimicrobial activity of the fullerene C60 and three types of fullerenols. C60(OH)12, C60(OH)36. 8H_2_0, and C60(OH)44. 8H_2_O were evaluated, and it was reported that although the pristine C60 did not manage to kill microorganisms, the fullerenols showed good antimicrobial activity against *P. acnes*, *S. epidermidis*, *C. albicans*, and *M. furfur* [43]. Although the preferred final application refers to anti-acne or anti-dandruff cosmetics, this study indicates that they could be tested as preservatives for the final application, especially C60(OH)44. 8H_2_O, which showed broad-spectrum antimicrobial activity.

#### 2.3.1. Graphene-Based NMs

Among the various allotropes of carbon, graphene, a planar graphitic sheet of graphite, presents the most biofunctional properties due to the ability to enrich its surface with several functional groups.

Graphene-based NPs can damage bacterial cells in two ways: either via the production of ROS or by trapping microorganisms in its aggregated sheets [44,45]. The main advantage of graphene oxide when used in packaging is that it can increase packaging hardness and strength while decreasing its weight. For example, carbon nanotubes were incorporated into a film composed of chitosan and polylactic acid for strawberry preservation. Results show that the film had low permeability and increased tensile strength and demonstrated antimicrobial efficacy [30]. In another study, graphene oxide nanosheets and clove essential oil were incorporated in a polylactic acid-based film and researchers proved its activity against Gram-positive and Gram-negative bacteria, and oxygen permeability also decreased due to the planar configuration of nanosheets [31]. Graphene can additionally be applied for the coating of the antibacterial agent iron oxide, and with the use of chitosan, produce a nanocomposite useful in both the biomedical and food industries [32]. Hemolysis tests indicated no toxicity of this nanocomposite. Due to safety arguments regarding the extended use of preservatives, the possibility to have a preservative in packaging films not in contact with the product but available on demand drove researchers to study the introduction of the phenolic compound salicylaldehyde to graphene oxide platforms [33]. The successful release of the preservative stimulated by synthesized acids at the ripening stage led to the conclusion that these nano-composites can be applied for the preparation of eco-friendly “fruit switches”.

#### 2.3.2. Carbon Dots

The preparation of carbon dots can be achieved either using “bottom-up” or “top-down” methodologies. Interestingly, they can be extracted through a one-step physical method from carbon black, the coloring agent commonly used in cosmetics and listed in the catalogue of NMs used in the European cosmetics market [46].

The properties of carbon dots that make them valuable in most applications are the tunable photoluminescence, high quantum yield, low toxicity, small size, biocompatibility, and low-cost sources [47]. In cosmetics, carbon dots were recently proposed for hair-dying applications, taking advantage of the interactions between carbon dots and hair proteins. On this assumption, various carbon dot species were doped on hair during dying with commercial colorants. Three species of carbon dots were found to make hair glow under 365 nm light, and those were the ones with the lowest absolute zeta potential [48].

Although the photoluminescent mechanism has not been fully investigated, this study proves their potential application both in hair and nail colorants. In addition, some recent studies investigate the introduction of carbon dots in sun care products as promising non-toxic, broad-spectrum, and eco-friendly ultraviolet absorbers [49]. In the food industry, they are currently used for the detection of food quality and safety, usually using fluorescence recovery or fluorescence quenching as detection principles to detect quality-related biomolecules such as ascorbic acid, tannic acid, and melamine in fruit juices, wine, and milk, respectively, or microorganisms and their metabolites such as *Salmonella* Typhimurium and aflatoxin B1 [50]. Due to their properties, carbon dots can be adsorbed on the bacterial cell surface and induce ROS production when exposed to light. This gives them antimicrobial characteristics. In a recent study, Kousheh et al. introduced carbon dots in a cellulose matrix and prepared a nanopaper that offered both antimicrobial and ultraviolet protection [51]. Carbon dots were produced from beneficial bacteria biomass, constituting a beneficial green approach. Generally, carbon dots can be produced from natural sources, including plant extracts rich in phytochemicals and fructose. Very recently, researchers prepared a nanopaper based on bacterial cellulose doped with carbon dots and supported its potential application in food preservation [52]. In addition, both studies reinforce the production of carbon dots from natural sources not only due to environmental concerns (green approach), but because of the high cost of using organic material as a precursor. Till now, and to the best of our knowledge, there are no studies reporting applications of carbon dots for the preservation of cosmetics.

#### 2.3.3. Carbon Nanotubes

Carbon nanotubes are cylindrical molecules that consist of rolled-up sheets of single-layer carbon atoms. They can be single-walled, with a diameter of less than 1 nanometer (nm), or multi-walled, consisting of several concentrically interlinked nanotubes with diameters reaching more than 100 nm. Nanotubes have 8 nm-diameter cavities that can encapsulate various functional materials in the food or cosmetic articles. A couple of patents have been filed in the field of hair colorants and cosmetic compositions [53,54]. Single-walled nanotubes have also displayed bactericidal activity against both Gram-positive and Gram-negative bacteria, but not much work has been done in this direction. Their microbial toxicity is mainly due to the oxidative stress that interrupts the continuation of the cell membrane or due to their adhesion to the microbial surface [55]. The application of carbon nanotubes as nanofillers in gelatin films has been successfully demonstrated, showing good results regarding the improvement of tensile strength and mechanical, thermal, and antimicrobial properties [56]. For instance, multiwalled carbon nanotubes were incorporated in a composite based on polylactic acid and chitosan to extend the shelf life of strawberries [30]. These researchers added carbon nanotubes in the composite, knowing that chitosan lags behind in heat stability and mechanical properties, and suggested a way to overcome these deficiencies.

Another type of carbon nanotube, called Halloysite nanotubes, have been introduced in nanocomposites to develop functional packaging for the extension of the shelf life of food and cosmetic products. Halloysite nanotubes are naturally occurring tubular clay NMs made of aluminosilicate kaolin sheets rolled several times. Biomaterials can interact with their surface due to aluminol and siloxane groups and form hydrogen bonds. Clay nanotubes have been studied for the deposition of hair dyes or anti-hair-loss active ingredients on hair [57], and have recently been used in nail polish and nail care products. The effectiveness of some essential oils as preservatives has already been mentioned, but their incorporation into active packaging has important drawbacks. High temperatures required for the formation of packaging can degrade such sensitive compounds and make them lose functionality. Halloysite nanotubes can protect the active molecules, keeping their quality and functionality in food-packaging systems [29]. Researchers studied the entrapment of carvacrol and thymol within halloysite nanotubes and concluded that using this method, the sensitive bioactive compounds retain their antimicrobial properties even in temperatures up to 250 °C [35]. The synergistic effect of the two essential oil components to preserve hummus spread was well demonstrated. Interestingly, the release of the two compounds within the product did not alter the organoleptic characteristics of hummus spread. Halloysites can be loaded both with liquid- and solid-phase ingredients and manage to slow down the release rate of those ingredients either by the admix of small NPs to make the end of the tube narrower or by coating the halloysite with higher-molecular-weight polymers like chitosan or gelatin in order to provide an additional barrier for the released ingredient [58]. In cosmetics, they have been applied to control the release of glycerol and solid sunscreen filters with success. Although no applications have been reported for the delivery of preservatives in a cosmetic product, it is expected that they could be a promising delivery system for them and a way to minimize their toxicity if their sustained release within the product for a long period is managed.

### 2.4. Nanoclays

Nanoclays are nanosize fluky soft silicate particles with a characteristic platelet form, with nanopores. These clays can be classified into four major groups: the kaolinite group from zeolite or halloysite, the montmorillonite/smectite group, the illite group, and the chlorite group. The montmorillonite group has gained more attention in the packaging sector due to the advantage of a high surface area with a large aspect ratio (50–1000) and good compatibility with most of the organic thermoplastics [59]. Many nanoclay materials are commercially focused on the development of improved food and cosmetics packaging [60,61,62]. They can be manufactured in many forms, such as rigid containers for beverages andflexible films for bread [63]. The incorporation of nanoclays into polymers depends on the type of polymer, the process, and the application, which determines packaging attributes, and these nanocomposites give unique mechanical and barrier properties to the container [62].

Except for improving packaging properties, nanoclays can be used as functional materials for active and intelligent packaging. They can provide antimicrobial or controlled release activity to the system and in the case of intelligent packaging, they can be used as colorimetric indicators. Several reports were found in the literature that demonstrated their antimicrobial function. For example, the addition of montmorillonite nanoclay in chitosan films applied on Gouda cheese increased their antimicrobial activity against *Escherichia coli*, *Staphylococcus aureus*, molds, and yeasts [64]. The minimum antimicrobial activity was observed for molds and yeasts. It is observed that this type of nanoclay is more effective against Gram-positive bacteria than Gram-negative ones. An explanation for this is the fact that Gram-positive bacterial envelopes are less complicated and the cationic groups of nanoclay, ammonium, or pyridinium, which are responsible for the antibacterial activity, bind on cells surface more easily.

The ability of nanoclays to entrap active ingredients and slowly release them into a product makes them valuable delivery systems for antimicrobial agents for the preservation of products on demand. Nanocomposites loaded with *Salvia macrosiphon* seed mucilage, *Rosmarinus officinalis* extract, carvacrol, potassium sorbate, and grapefruit seed extract were found to improve the antibacterial effect of active packaging, while at the same time maintaining food quality [65,66,67].

### 2.5. Metallic NPs and Their Nanocomposites

Metallic NPs, such as silver NPs, titanium dioxide NPs, and zinc oxide NPs, are extensively applied to make nanocomposites due to their antimicrobial properties, and most of their applications are found in the food industry. Bhuyan et al. presented the various antimicrobial mechanisms of metallic NPs [68]. Silver NPs release silver ions (Ag^+^) that cling to the microbial cell surface and disorganize the cell membrane, leading to the destruction of DNA and ending in cell death. More specifically, the uptake of free Ag^+^ ions forces the production of ATP, and in combination with the disruption of the DNA replication, ends up creating reactive oxygen species (ROS) [69]. Regarding the mode of action of titanium NPs, this is triggered through exposure to UV irradiation, which causes the production of ROS, leading to the lipid peroxidation of phospholipids in the cell membranes and bacterial death [70].

This photocatalytic activity as the starting point of titanium NP activation is the major handicap in the selection of this nanoparticle as a preservative. Even though zinc NPs possess high photocatalytic efficiency among all inorganic photocatalytic materials, it has been observed that the antibacterial activity of ZnO can be demonstrated likewise in the dark, inhibiting the bacterial growth [71].

#### 2.5.1. Silver NPs

Concerning cosmetics, among the registered nano-ingredients in Europe, those that claim antimicrobial properties are NPs based on colloidal gold, platinum, and silver. The term “colloidal” implies that the particles are in aqueous suspension. In spite of their claimed antimicrobial activity, none of these metallic NPs have been approved as preservatives. Nano-colloidal gold and platinum suspensions are currently used in antiaging products to stimulate cell turnover and natural healing. This ability is based on the property of promoting electron transfer between metal ions naturally found in the skin. About 19 trade ingredients of colloidal gold and eight of colloidal platinum are registered [72,73]. None of them are listed in the preservatives list. Nano-colloidal silver is intended to be used as an antimicrobial agent in cosmetics, including toothpastes and skin care products, with a maximum concentration limit of 1%. It is commonly used in cosmetic products with bactericidal effects. It is well known that silver ions and silver-based compounds have strong antimicrobial effects [74,75]. However, these silver-based compounds gradually precipitate in solutions, resulting in the diminishing of their efficacy. On the other hand, silver NPs do not present these limitations and are more effective as cosmetic preservatives. The reduction of the particle size results in more stable silver solutions, with higher efficiency at killing both bacteria and fungi without penetrating human skin [76]. A recent comparative study between silver and gold NPs reported differences in the structure of the skin care product where they were applied. More specifically, silver NPs created agglomerates, whereas gold NPs did not [77]. In the same study, it was also reported that the fungicidal properties against *A. niger* and *S. cerevisiae* of both NPs were different. Sensory profile, smell, and color assessments of the tested creams demonstrated that the 200 mg/kg gold nanoparticle cream had a better performance.

Silver NPs and their nanocomposites are among the most widely used NMs in the food industry [78]. Silver has been authorized by the European Food Safety Authority (EFSA) as food additive for coloring food (E174). According to the EFSA, approximately 20% of the E174 used is from confectionery pearls. This is the reason why Narciso et al. conducted a study to investigate the accumulation of silver NPs in the gastrointestinal system and its possible toxicological effect. According to this study, silver NPs did not cause tissue damage or genotoxicity even though it was accumulated on the emptied duodenum due to the fact that they never passed into the cell nucleus [79]. Colloidal silver is still banned as food ingredient. On the other hand, colloidal silver proteins are consumed in the USA as a functional food. The bacteriostatic and bactericidal concentrations of electrically generated silver were determined in 1976 [74]. Since silver is not approved for direct addition to food, most of its applications are found in food packaging. A recent study demonstrated that a polyethylene composite film containing nanosilver showed great potential in developing an antibacterial and acidic food packaging system, while at the same time enhancing its barrier properties [80]. Other applications using silver NPs in food packaging and films are presented in Table 2. As can be seen, this kind of technology finds applications in a wide range of food, such as raw chicken fillets, bread, and nuts. 

Instead of using silver NPs as preservatives alone, a wise application is to boost their activity with the combination of conventional preservatives by conjugation. For example, a recent study applied conjugated silver NPs on the surface of sodium benzoate (E211) and created a stable antimicrobial composite with significant efficiency against the food-borne pathogens *Salmonella* typhimurium type 2, Shiga toxin-producing *E. coli* (STEC), *B. cereus*, and *S. aureus* [92]. Given the fact that sodium benzoate-functionalized silver NPs were produced in water and the requested amount of sodium benzoate was reduced, this application offers a green and probably safer alternative for food preservation.

#### 2.5.2. Titanium Dioxide NPs

Both titanium and zinc NPs are extensively used in the cosmetics industry, but not for preservation reasons. Titanium dioxide is applied mainly in sunscreens for ultraviolet protection and in color cosmetics to neutralize color pigments. Even though it does not seem to present any health risk, the Scientific Committee on Consumer Safety (SCCS) of the European Commission does not recommend its use in sprayable formulations. In the food industry, titanium dioxide is no longer considered safe when used as a food additive. Due to its ability to scatter visible light, it was added in food formulations to make products look white and bright. As already mentioned, the mode of action of titanium NPs is triggered from the exposure to UV irradiation, which causes the production of ROS, leading to the lipid peroxidation of phospholipids in the cell membranes of the bacteria and their subsequent death.

Titanium NPs are usually among the key ingredients constructing multilayered nanocomposites added in active packaging to extend the shelf life of specific food products. A nanocomposite film consisting of glycerol, cellulose nanocrystals, TiO_2_ NPs, and wheat gluten was coated over a kraft paper in three layers and was found to exhibit excellent antimicrobial activities against *S. cerevisiae*, *E. coli*, and *S. aureus* after 2 h of exposure to UVA light illumination [93]. In another study, a shelf-life extension of 6 days was observed following the application of a nanocomposite film composed of TiO_2_ NPs and rosemary essential oil on lamb meat during storage under refrigeration [94]. Similarly, a packaging system made of silver and titanium nanocomposite was able to significantly extend the shelf life and improve the microbiological safety of bread in comparison with bread packed in high-density polyethylene (HDP) and bread not subject to packaging [82].

#### 2.5.3. Zinc Oxide NPs

Zinc oxide has been considered a valuable active ingredient for the treatment of various skin disorders since ancient times, and it is the key ingredient in diaper rash ointments. In addition, zinc NPs are the broadest spectrum UVA and UVB filters approved for use by European and USA authorities. Being a promising preservative for the pharma and cosmetics industries, zinc oxide has been studied in various sizes and concentrations against different microorganisms. A recent study explained the effect of the size of zinc NPs in their antimicrobial efficacy and demonstrated that this increases by decreasing their size [95]. This activity can be enhanced by γ irradiation maybe because of the effect on the nanoparticle diameter after irradiation [96].

Zinc NPs display high migration phenomena when they are in contact with an acidic environment. This was shown in a recent study where a homogeneous dispersion of such NPs was incorporated into a starch-based flexible coating for food-packaging paper [97]. Even though migration is a negative issue, whether it is within legislative limits can be measured and confirmed [98].

### 2.6. Nanoencapsulation and Delivery Systems for Preservatives

Encapsulation is the entrapment of molecules within a system and aims to protect them from the effects of environmental factors without losing any of its properties. Encapsulation can be promoted in micro or nanoscale. Nanoencapsulation in particular offers the advantages of increasing the solubility of some specific molecules and promoting their slow release. This can be exploited for the delivery of antimicrobial agents for the preservation of cosmetic and food products through the production of nanoemulsions, nanoliposomes, lipid carriers, nanofibers, and polymeric NMs.

#### 2.6.1. Nanoemulsions

Nanoemulsions are oil-in-water or water-in-oil colloidal dispersions where the diameter of the droplets of the inner phase ranges from between a few nanometers to 200 nm [99]. This small droplet size gives them unique properties, including enhanced solubility, optical transparency, stability against sedimentation, and creaming for the delivery of a wide range of bioactive compounds. In comparison to macroemulsions, they are kinetically stable. Their preparation method requires high or low energy, and the final polydispersity index is typically low. Surfactant concentration and length as well as ultrasonication time control the final droplet size. Nanoemulsions are used in the food industry to create processed dressings. The use of nanoemulsions as antimicrobial agents is a very promising innovation. Their mode of action relies on the electrostatic attraction between the cationic charge of the emulsion and the anionic charge of the bacterial membrane. Due to the attractive forces, nanoemulsion particles fuse with lipids of the bacterial cell membrane, and when enough particles are fused, a part of the trapped emulsion energy is released, resulting in cell lysis.

Several companies of raw materials have used nanoemulsions as delivery systems to preserve bioactive compounds. For instance, vitamins, plant extracts, and essential oils are facing oxidation or solubilization problems and need a vehicle to deliver them effectively on skin, increasing their bioavailability. The replacement of synthetic preservatives with natural alternatives is a growing demand in both the cosmetics and food industries. Essential oils contain terpenes, terpenoids, phenylpropanoids, and other molecules that demonstrate antimicrobial properties. For instance, a comparative study between essential oils of *Lavandulla officinallis*, *Melaleuca alternifolia*, and *Cinnamomum zeylanicum* with methylparaben, which is used as a cosmetic preservative, showed that all those essential oils could replace the synthetic preservative [100]. Another study investigated specific essential oils when combined with common cosmetic preservatives and in certain cosmetic preparations, and showed that the essential oils increased the effectiveness of the preservatives against *P. aeruginosa* and *S. aureus* [101]. The incorporation of essential oils as preservatives in cosmetics or food products has limitations, especially regarding their volatile aromatic compounds, which may give an undesirable smell and taste to the final product. Because of such limitations, essential oils cannot be added in formulations in high concentrations, leading to a loss of effectiveness. The entrapment of essential oils or their bioactive compounds in nanoemulsions to improve their preservation function has found applications in many different products, and some representative examples are presented in Table 3.

Tween 80 is the emulsifier used in most of the studies, and formulations vary depending on the application. The oil phase composition of nanoemulsions, ripening inhibitor type, and concentration can influence the antimicrobial activity of the essential oils [117]. The main destabilization process of such nanoemulsions is the Ostwald ripening phenomenon, where an increase in the oil droplet size is promoted mainly in the first 24 h after production. One of the proposed strategies to overcome such aging is the addition of a gelling agent or a gum into the dispersed phase. Aggregation is another phenomenon that can occur during the lifetime of a nanoemulsion. Polymer-coated nanoemulsions can delay the onset of this phenomenon, and recent studies suggest the use of the cationic biopolymer chitosan due to the generation of both electrostatic and steric repulsive interactions [118].

#### 2.6.2. Nanoliposomes

Nanoliposomes are among the most investigated nanocarriers in the cosmetics and food industries. They are vesicular systems with an aqueous core and are surrounded by a lipophilic bilayer, offering the advantage to carry and deliver both hydrophilic and hydrophobic molecules. Nanoliposomes are produced through the assembly of amphipathic molecules, usually phospholipids, and the techniques of their preparation are divided into passive and active loading. In the cosmetics industry, liposomes find applications as delivery vehicles for various bioactive compounds such as vitamins, peptides, phytosterols, and phytocompounds. The encapsulation of cosmetic preservatives into liposomes has not yet been studied, but several studies demonstrate the benefits of using nanoliposomes as delivery systems of antimicrobial compounds not yet registered as preservatives. A recent such study investigated three essential oils distilled from *Artemisia afra*, *Eucalyptus globulus*, and *Melaleuca alternifolia* encapsulated in nanoliposomes based on diastearoyl phosphatidylcholine and diastearoyl phosphatidylethanolamine and tested them for their antimicrobial efficacy [119]. The *E. globulus* and *M. alternifolia* liposomes showed the lowest minimum inhibitory concentrations, but further coating with polymers improved their stability.

In contrast to the fate of nanoliposomes in the cosmetics industry, nanoliposomal formulations delivering antimicrobial agents have found many applications in the food industry. Some of their most promising applications are presented in Table 4, and many of them could have a potential use in cosmetics as well.

It is well noticed that the entrapment of nanoliposomes delivering preservatives in edible films minimizes their antimicrobial activity due to the inhibition of their release from the matrix [123]. Another reason why nanoliposomal formulations can diminish the preservation activity of some molecules is the negative charge of the zeta potential. Thus, the electrostatic repulsion between the negatively charged nanoliposomes and the negatively charged bacteria results in lower interaction between the encapsulated preservative and the bacteria, and as a consequence, less antibacterial activity [129]. On the other hand, the surface modification of liposomes could improve the stability of the liposomal membrane while at the same time avoid lipid oxidation that can often affect liposomes [124]

#### 2.6.3. Niosomes

Niosomes can be mentioned as an advanced version of liposomes. They are delivery vehicles with a closed bilayer structure composed of non-ionic surfactants. They are recognized as safer and cheaper than liposomes and more stable, with a longer self-life [130,131]. Nanoliposomes are prone to oxidation due to sensitive phospholipids [132]. On the other hand, niosomes face leakage deficiencies, a phenomenon that can be corrected by the addition of cholesterol [133]. Applications of niosomes can be found in both cosmetic and pharmaceutical industries, taking advantage of their property of enhancing the bioavailability of bioactive compounds [134]. They are usually applied for the protection of sensitive compounds such as vitamin A and Ε or resveratrol [134,135,136]. Interestingly, niosomes provide a protective delivery system for antibiotics, making them remarkably effective in orthopedic, orthodontic, ophthalmological, and other treatments [137]. Applications in the food industry are very few, and their use in a preservation system has not been studied yet.

#### 2.6.4. Solid Lipid NPs and Nanostructured Lipid Carriers

Solid lipid NPs (SLNs) are NPs composed of lipids with a solid lipid matrix. Their nanometer size offers them unique properties such as high drug-loading capacity and long-term stability. Their production does not require organic solvents; therefore, they can support green chemistry claims. Nanostructured lipid carriers (NLCs) belong to the second-generation lipid NPs and are the result of the combination of solid and liquid lipids. In comparison to SLNs, NLCs have a distorted structure, with spaces for the accommodation of biomolecules, and offer better loading capacity and stability. In the last 15 years, SLNs and NLCs have been the most common carriers of active ingredients used in the cosmetics and food industries and arose from the need to overcome the deficiencies of liposomes, nanoemulsions, and polymeric NPs [129]. Some of the current applications of lipid NPs in these industries are summarized in Table 5.

Lipid nanoparticle applications aim to protect the transported biomolecule and increase its bioavailability. Many different formulations have been tested to create the ideal SLN or NLC, and depending on the biomolecule, the target, or the type of the final product, different lipids make up the nanoparticle. In addition to that, several manufacturing procedures are applied for each formulation, such as double emulsion and melt dispersion [154], high pressure homogenization cold dispersion [155], high pressure homogenization hot dispersion [156], warm microemulsion [157], supercritical fluid [158], and solvent displacement [159]. Even though there is a plethora of different formulations and manufacturing processes for SLNs, scientists are still investigating new cost-effective and scalable production methods [160]. SLNs and NLCs applied for the preservation of cosmetics and food are summarized in Table 6.

SLNs made with pure precirol and NLCs made of a mixture of precirol and almond oil are able to act as reservoir systems for parabens and their mixtures [153]. The sustained released of parabens minimizes product contamination during usage by the consumer, and at the same time reduces the toxicity of such preservatives in cosmetics. The prolonged antibacterial activity of nisin against *L. monocytogenes* and *L. plantarum*, when encapsulated in SLNs, was evident for more than 15 days in comparison to free nisin [161]. In that case, SLNs were formulated with the emulsifier glyceryl monostearate and the main surfactant poloxamer 188. Their manufacturing process was based on hot high-pressure homogenization. Interestingly, in another study, SLNs loaded with carvacrol were more effective against food spoilage bacteria when fabricated with propylene glycol monopalmitate and glyceryl monostearate in a ratio of 1:1 [152].

#### 2.6.5. Polymeric NPs

Polymers are widely used for the delivery of preservatives either in the form of NPs or in more complicated forms for the stabilization of other delivery systems. A polymeric nanoparticle is a particle within the size range of 1 to 1000 nm with an oily or hydrophilic core that is surrounded by a polymeric substance or a polymeric core surrounded by an adsorbed substance. Polymeric NPs may have two different structures composing nanocapsules and nanospheres [162]. Biocompatible polymers, such as hyaluronic acid, are preferred in biomedicine to manage drug or gene delivery and stem cell microencapsulation for tissue regeneration or repair [163].

In cosmetics, this type of NP is used for the protection and delivery of sensitive t compounds that bear low biocompatibility, the delivery of aromatic compounds, or to mask the odor of some chemicals. In most of the applications, especially in food packaging, metallic NPs are not used alone but in combination with other materials such as nanoclays or polymers [164]. This is a way to decrease their toxicity or to increase their stability and promote prolonged protection. For example, silver NPs are not stable and tend to agglomerate, resulting in the reduction of their antimicrobial properties. A successful combination was achieved between carrageenan-based nanocomposites and laponite, a synthetic clay with unique properties [165]. In that study, the polysaccharide carrageenan was chosen among other polymers due to its eco-friendly profile and strong hydrogel properties. More specifically, carrageenan hydrogels show increased water-holding capacity and strong barrier function, and their combination with clays results in adhesive and strong composites with improved properties. It is suggested that chitosan, a cationic linear polysaccharide, and alginate, an anionic polysaccharide when blended to perform a coating for nanoliposomes—so-called “colloidosome”—can overcome their stability limitations [121]. Similarly, pectin and polygalacturonic acid-coated liposomes have been studied for the effective delivery of the polypeptide nisin and successfully lowered the release rate of this preservative, offering a safer delivery of it to the product [166].

#### 2.6.6. Nanofibers

Nanofibers are NMs with a diameter in the nanometer range and can be produced by polymers using the electrospinning technique. This applies electric force to draw charged loads from polymer solutions to produce ultrafine fibers. Depending on the desired application and the properties required, electrospinning parameters, being the electric field applied, flow rate, needle diameter, environmental conditions, and solution characteristics, are alternated. Electrospun nanofibers find applications in several industries due to their unique properties and mainly because different compounds can be added in the polymer solution and thus be trapped in the nanofiber. Nanofibers for medical and cosmetics market size are forecast to reach USD 4.2 billion by 2026 [167]. In cosmetics, nanofibers find applications as delivery systems for active ingredients and for face masks, and the most commonly used polymer is polyester polycaprolactone due to both its biocompatibility and biodegradability.

Regarding food preservation, the encapsulation of antimicrobial agents in nanofibers has been studied, and results showed that they can be applied for the preservation of different food products such as cheese, processed meat, fresh juices, and yogurt mainly as a component of the packaging material [168,169,170].

Thus, the encapsulation of antimicrobials in nanofibers promotes their slow release in the product, increasing their bioavailability and action. For instance, the preservative nisin was successfully loaded in nanofibers made of poly-g-glutamic acid and chitosan, achieving the slow release of this preservative in cheese against *Listeria monocytogenes* [168]. Nisin, when applied directly in the product, can interact with other components and lose part of its preservation activity. On the other hand, when it is delivered in a protected environment, its bioavailability is increased. This was also demonstrated in a comparative study between free nisin and nisin-loaded chitosan/alginate NPs, where encapsulated nisin gave better results regarding the preservation of Greek feta cheese, extending its shelf life without affecting its taste [169].

Similarly, ginger essential oil was loaded in a polymeric blend composed of soy protein, polyethylene oxide, and zein in a ratio of 1:1:1 (*v*/*v*/*v*), and its antimicrobial activity against *L. monocytogenes* was tested in situ in fresh Minas cheese [170]. After the third day of storage, the bacterial counts had decreased, and at the ninth day the counts had decreased from 4.39 log CFU·g^−1^ to 3.62 log CFU·g^−1^ (*p* < 0.05), presenting a higher reduction compared with the stored samples without loaded nanofibers.

### 2.7. Nanofluids

A nanofluid is a colloidal suspension that contains NPs made from metals, carbon nanotubes, oxides, carbides, and others, and the base can be water, oil, or ethylene glycol. Their production can be achieved via one-step or two-step strategies where the main goal is to avoid sedimentation and aggregation phenomena. Nanofluids have thermophysical properties known as thermal conductivity, dynamic viscosity, density, and specific heat, which are influenced by their size and shape, as well as temperature and the concentration of the NPs in the suspension. In the food industry, the technology of nanofluids finds applications in processes that require heat exchange, known as freezing, dying, pasteurization, and others. For instance, thermal pasteurization is a technique commonly applied for the preservation of milk and fruit juices. This is a technique that efficiently extends the shelf life of such products but affects their nutritional quality and organoleptic characteristics due to the denaturation of proteins and degradation of bioactive compounds such as vitamins and other nutritional compounds. Thus, scientists have focused on the investigation of reducing heat time for such heat techniques to avoid their negative effects, introducing nanofluid thermal processing, which is based on the high thermal conductivity of nanofluids in comparison to other fluids. It was found that heat transfer and thermal conductivity increased when the particle size of the suspended NPs decreased. In another relevant study, scientists used a nanofluid composed of alumina NPs for the thermal processing of watermelon juice and found that the final product contained vitamin C and lycopene at 6% and 10% higher concentrations, respectively, compared to the product receiving conventional heating [164]. Other studies have introduced TiO_2_/water and alumina nanofluids in milk pasteurization, and all of them proposed nanofluids as an alternative to water for heat treatment preservation techniques [171,172].

### 2.8. Nanomilling

Nanomilling is a “top-down” process by which the industry can minimize on the nanoscale the size of a poorly water-soluble active ingredient, increasing its surface area and consequently increasing its rate of dissolution. This technique uses ceramic microbeads or those constructed from crosslinked polystyrene, while the processed compound is dispersed in an aqueous medium stabilized with a surfactant or polymer to avoid its aggregation. This process was recently applied in the pharma industry to generate nanoparticulate suspensions of hydrophobic active pharmaceutical ingredients (APIs) that are widely used, such us rapamycin, sirolimus, fenofibrate, tizanidine HCl, aripiprazole lauroxil, and rilpivirine [173]. The increase in the surface area of the drug after the nanomilling process tremendously improves its bioavailability.

Curcumin is a phenolic compound found in turmeric that possesses a plethora of activities, such as being antioxidant, anti-inflammatory, and anti-microbial. The main disadvantage of this molecule is its lipophilic structure, which effects solubility and bioavailability. For this reason, a study based on the reduction of the size of the molecule by wet nanomilling was conducted, and the effect of this reduction in nano range was determined concerning the increase of its antimicrobial activity [174]. Transmission electron microscopy was applied to determine the antibacterial activity, and interestingly, the images taken showed that nanocurcumin particles damaged the bacterial cell wall and penetrated into the cell, causing its death. The suspension was based in water and the microorganisms that were studied consisted of two Gram-negative bacterial species (*E. coli* and *P. aeruginosa*) and two fungal species (*P. notatum* and *A. niger*). In another study, nanomilling was applied in orange juice pulp for the preparation of orange juice [175]. A slight reduction in the content of ascorbic acid was observed, but at the same time the initial microbial load was decreased and the stability of the juice was enhanced. This sounds like a promising approach for future applications of curcumin as a preservative to extend the shelf life of water-based products; however, more studies need to be conducted concerning the potential toxicity of nanomilling products to human cells.

### 2.9. Nanozymes

Nanozymes (also called enzyme-mimetic NMs) combine the catalytic properties of enzymes with those given from their size in nanoscale. The use of natural enzymes has some drawbacks, such as the high cost of their purification, which requires a long amount of time, and their possible denaturation due to the high temperatures prevailing in harsh environmental conditions. Those drawbacks can be successfully addressed with the use of artificial nanozymes. Except for their use as recognition receptors, they are used as signal tags mainly in ELISA (enzyme-linked immunosorbent) and LFIA (lateral flow immunoassay) assays, as multifunctional sensing elements when modified with other molecules, or for signal amplification in food-detection assays, working synergistically with other NMs [176].

Nanozymes are not currently applied as preservatives in food or cosmetics. However, their functions may contribute to the improvement of the current preservation strategies since they can detect undesired molecules in a given product. More specifically, they can be applied to detect excessive use of some preservatives such as hydrogen peroxide (H_2_O_2_) in milk [177], harmful ions [178], antibiotics in meat products [179], organophosphate pesticides [180], mycotoxins [181] and food-borne pathogens such as *E. coli*, *S.* Enteritidis, and *V. parahaemolyticus* in sea food [182].

## 3. Discussion

Even though the application of NMs in the food and cosmetic industries is still in its infancy, dozens of reports demonstrate the advantage of using nanotechnology for the preservation of cosmetics and food, primarily in the field of food packaging. A wide range of nanostructured materials, from metallic NPs and their nanocomposites to nano-inspired delivery systems for natural preservatives, has been applied in both industries, transforming many domains of scientific knowledge so far in those fields. For instance, silver NPs are expected to play an important role in designing polymeric materials used in the field of active food packaging, while at the same time bioplastics are expected to replace non-biodegradable polymers, and with the support of nanocomposites, to increase the shelf life and improve the quality of the products. Bioplastics produced by renewable biomass sources, metallic NPs coming from bio-fermentation, and nanoclays as extracted minerals could be included in the components of sustainable smart food. The on-demand release of preservatives from active packaging eliminates harmful effects of chemical preservatives, while the entrapment of natural antimicrobial agents contributes to the consumption of more natural products. The replacement of water by nanofluids in heat treatment of fruit juices and other liquid foods can provide beverages with the highest quality, and nanomilling is a cost-effective way to enhance the preservation activity and bioavailability of several natural compounds. Food waste, one major issue with impacts in public health, the economy, and the environment, could also be managed with the application of nanotechnology. The hazard of product contamination will be minimized, and packaging waste will not harm the environment while improving product quality.

Further research is needed to understand better the molecular mode of action of several antimicrobial nanostructures. Similarly, more comparative studies between different formulations used for active packaging are desired since they could contribute to underlining the special characteristics of each nanomaterial. Furthermore, it has been noticed that most reports concern materials that are applied in lab scale, and the scale up could present unexpected difficulties. Even though this scale up can be efficiently managed, the cost effectiveness of the new technology for the needs of the industry should be examined.

In the framework of this review article, it was noted that very few articles have been published regarding the applications of nanotechnology in cosmetics for preservation purposes in contrast to those published concerning the food preservation sector. At the same time, however, most of the reports demonstrating the application of nanotechnology in active antimicrobial food packaging claim that their results could also be exploited to support cosmetic applications. Nevertheless, it should not be neglected that cosmetics preservation targets different detrimental bacteria than those spoiling food, and cosmetics shelf life is usually estimated at two or three years, which is a much longer period compared to the shelf life of most food products. In addition, the repeated use of a face cream, for instance, makes contamination of the product by the consumer more likely, whereas most of the processed food products have a short secondary life after the opening of the package.

The scientific community and the relevant stakeholders have recognized the significant contribution of nanotechnology to the food and cosmetics industries, including its great potential for preservation purposes. In this direction, various projects have been funded recently, targeting the development of active packaging solutions driven by nanotechnology. In one such project funded by the European commission, called “Acticospack” [183], the scope was to develop active packaging solutions for three target cosmetic products aiming to reduce their content in preservatives, while at the same time keeping the quality and safety of these products through the same or even longer shelf life [180]. The project successfully ended in 2017 and led to the development of an active PET bottle for shampoo, an active PE bottle for a sun care product, and an active PP pot for a day cream, all capable of extending the shelf life of the products and minimizing the use of chemical preservatives by 25 to 40%.

In the same direction, the European Union’s Horizon 2020 Research and Innovation Programme funded the implementation of the Nanopack program (starting in 2017 with a three-year duration) for the creation of novel smart and eco-friendly antimicrobial packaging solutions that minimize the required preservative concentration in bulk food products, while at the same time meeting legal, regulatory, safety, and environmental requirements [184]. For this project, the bread, dairy, and meat industries; research institutes; and universities collaborated with the aim of creating smart packaging to extend the life of products while protecting consumer safety and minimizing food waste [185]. In the framework of this project, natural haloysite nanotubes were introduced in a biopolymeric matrix and were loaded with essential oil compounds of geraniol, linalool, menthol, thymol, carvacrol, and citral, which have antimicrobial properties. The compounds were slowly released from the packaging onto the food surface, minimizing microbial spoilage. The main goal of this study was to minimize food waste and offer eco-friendly, safe packaging to the community. Simon van Dam, a senior member of the Nanopack project, said that the developed technology could be applied to cosmetics packaging as well [186].

One of the main obstacles currently limiting the expansion of nanostructured materials in the industry is their possible toxicological and eco-toxicological effect on human and animal health, respectively. The exposure routes of NPs are inhalation, skin penetration, and digestion. NPs should be treated as new chemicals requiring new safety assessments before being allowed for use in any consumer product. On 8 October 2020, the European parliament announced the ban of titanium dioxide, referred to as E171, as a food additive because in many foods the proportions of NPs were greater than 50% and not labeled as “nano” [175]. This ban does not refer to the titanium dioxide added in the packaging material. Thus, the use of NPs as food ingredients is considered potentially more harmful than their application in food packaging. However, the release of them within the product should still be measured. Although authorities have published testing methodologies and specific limits for the migration of NPs, these do not fit for all possible applications. In addition to that, a harmonized regulatory policy for nanostructured materials and their possible harmful effects to humans and the environment has not been yet established, which would result in increased consumer awareness. In this direction, growing demand has been raised to establish internationally granted protocols to determine exposure assessments, the toxicokinetics, and the toxicity of NMs, including those with applications in cosmetics and food.

Although the European Union essentially relies on the relevant legislation on food packaging and acknowledges that once a packaging material has been accepted for food, it is more easily approved for cosmetics [187], it should be noted that the two sectors should not be harmonized in such a way since the exposure route is different. Once a nanoparticle enters the gastrointestinal tract, several physicochemical interactions can occur with other contents of the system that could affect nanoparticle properties. On the other hand, one might reasonably wonder whether the concentration of a preservative that is actually bactericidal could also be lethal for skin cells. Several studies have investigated the possible toxicity of NPs to human cells depending on the route of exposure, but each one concerns specific applications, and as thus its conclusions cannot always be generalized mainly due to the extreme diversity of other chemicals that coexist in the testing formula. In addition, conflicting results often make it difficult to evaluate the real impact on human health. Last but not least, most of the studies fail to represent the human organism well. For instance, a recent review on the impact of inorganic NPs on the human gut microbiome criticized that the evaluation of microbial communities is usually done in mouse models, despite the fact that 85% of the 16S rRNA sequences of mouse microbiota represents bacterial genera that do not exist in humans [188,189]. For that reason, scientists suggest the pro-inoculation of the mice gastrointestinal system with human gut microbiota when investigating for possible alterations in microbial populations.

The prospect of eliminating the possible toxicity of some NPs by coating them with polymers has been discussed frequently, and many reports demonstrate the effectiveness of this application. This is indeed a technique that is well recognized in the medical sector, where biopolymers are commercially used for the improvement of NP biocompatibility. However, in the case of using NPs as preservatives, the importance of maintaining the antimicrobial property of these coated NPs should be taken into consideration. For instance, silver NPs are considered toxic to mammalian cells, causing an inflammatory response and inducing cytotoxicity and genotoxicity in human cells. Interestingly, it has been reported that coating them with a chitosan polymer matrix can reduce their toxic effects, at the same time maintaining their bactericidal activity [190].

Finally, the possible multifunctionality of the nano-preservative system is a factor that should be taken into consideration when designing contemporary foods or cosmetics. Ideal nano-ingredients could be designed to bear both efficacy for the bulk, acting as preservatives, and for the user, acting as bioactive ingredients. Indeed, antioxidant molecules formulated into rationally designed nano-forms could, for example, scavenge free radicals that deteriorate the quality of the formulation and act as nano-antioxidants that offer protection to the user from environmental aggressors [191].

## 4. Conclusions

The present review presented the involvement of nanotechnology in the evolution of food and cosmetic preservation. As it turns out, science has already made a significant contribution to this field, and rapid developments in the last few years reinforce the belief that in the future much of the preservation strategies to be pursued by the two industries will be based on NPs and their nanocomposites. By doing this, many harmful chemical preservatives could be replaced by nanostructures, and at the same time the efficiency of natural antimicrobials could be further enhanced by their encapsulation and on-demand release once the product is contaminated. Surely, the role of packaging is crucial to extending the shelf life of products. The possibilities offered by the science of nanotechnology are enormous, and nanocomposites could be a way to minimize the reckless use of preservatives in products, while at the same time taking care of the environment and enhancing the mechanical and physical properties of bioplastics. The rapid technological development in this growing sector makes it almost impossible for the authorities to respond immediately, as new protocols for toxicity and migration testing have to be established in order to apply new protective limits for the consumer and assure the eco-safety of all new applications. Funding should continue providing resources for comparative studies between old and novel applications, and most importantly, case-by-case assessments, migration, and toxicological studies. Last but not least, the vision of smart food and cosmetics packaging should be enhanced with consumer education on all these novel assistive technologies and their advantages.

## Figures and Tables

**Table 1 nanomaterials-12-01196-t001:** Applications of carbon-based nanofillers for cosmetics and food preservation.

Composition	Carbon-Based Nanomaterial	Target Microorganism	Application	References
Polylactic acid (PLA)/carbon nanotubes (CNTs)/chitosan (CS) composite fibers	Carbon nanotubes	*Staphylococcus aureus*, *Escherichia coli*, *Botrytis cinerea* and *Rhizopus* spp.	Strawberries	[30]
Polylactide/graphene oxide nanosheets/clove essential oil	Graphene oxide sheets	*Staphylococcus aureus* and *Escherichia coli*	n/a	[31]
Chitosan–iron oxide nano-composite hydrogel	Iron oxide-coated graphene oxide	Methicillin-resistant *Staphylococcus aureus*, *Staphylococcus aureus*, and *Escherichia coli*, as well as opportunistic dermatophyte *Candida albicans*	n/a	[32]
Paper	Graphene oxide platforms	*Pseudomonas syringae*, *Escherichia coli*	Eco-friendly fruit switches	[33]
Nanocellulose matrix	Carbon dots	*Escherichia coli*, and *Listeria monocytogenes*	Films	[34]
Low-density polyethylene	Halloysite nanotubes	*Escherichia coli*	Films for hummus spread	[35]

**Table 2 nanomaterials-12-01196-t002:** Applications of silver NPs for food preservation.

Composite Production Method	Composition	Target Microorganism	References
Solid ion exchange	Silver–nanoclay (montmorillonite clay)	Gram-negative bacteria	[81]
Sol–gel procedure	Silver/TiO_2_ nanocomposite	*B. subtilis* and *B. cereus*	[82]
Extrusion	Silver NPs embedded in distinct carriers (silica and titanium dioxide) with low-density polyethilene	*E. coli* and *S. aureus*	[83]
Extrusion	Silver NPs in low-density polyethylene	Fungi and Gram-negative bacteria	[84]
Spray coating	Silver-coated low-density polyethylene films	*P. fluorescens*, *S. aureus* and microflora isolated from raw chicken	[85]
*In situ* melt blending method	Dodecyl mercaptan-functionalized silver NPs integrated with polypropylene nanocomposite	Gram-negative (*E. coli*) and Gram-positive (*S. aureus*)	[86]
Solution casting method	Silver NPs, pullulan, and pectin	*E. coli*, *L. monocytogenes*, *S. Typhimurium*, *S. aureus*, *B. cereus*	[87]
Solution casting method	Silver NPs and pectin	*E. coli* and *L. monocytogenes*	[88]
Laser ablation method	Silver NPs and agar	*L. monocytogenes* and *E. coli*	[89]
γ-ray irradiation	Silver NPs and poly(lactic acid)	*E. coli* and *S. aureus*	[90]
Extrusion	Polyethylene nano-silver composite films	Molds	[91]

**Table 3 nanomaterials-12-01196-t003:** Applications of nanoemulsions in the cosmetics and food industries as preservation systems.

Application	Essential Oils and Main Constituents	Target Microorganism	Nanoemulsion Formula Info	References
Fish-processing industry	Lemon essential oil (d-limonene, p-cymene, β-pinene)	Food-borne pathogens and fish spoilage bacteria (*P. damselae, E. faecalis, V. vulnificus, P. mirabilis, S. liquefaciens*, and *P. luteola*)	Tween 80 (1% *w*/*w*) and water (89% *w*/*w*), homogenized by using an ultrasonic homogenizer for 15 min at 72 amplitudes	[102]
Food, cosmetics, and agrochemical industries	Pure citral as a constituent essential oil from citrus fruits	*S. aureus, P. aeruginosa, E. faecalis, S. typhimurium*, and *L. monocytogenes*	Span 85 (sorbitane trioleate) and Brij 97 (polyoxyethylene (10) oleyl ether). Two-stage process (polytron and ultrasonic)	[103]
Cereal grains (wheat, barley, and corn)	Thyme oil (thymol, p-cymene, γ-terpinene, and linalool), lemongrass oil (β-citral, αcitral, D-limonene, and geraniol), peppermint oil (menthol, L-menthone, eucalyptol), cinnamol oil (eugenol), caryophyllene, benzyl benzoate, linalool), clove oil (eugenol, caryophyllene, α-humulene, eugenol acetate)	Two isolates of *F*. *graminearum*	0.5 wt% Tween 80, 5 wt% of total oil phase, and 94.5 wt% phosphate buffer solution (10 mM, pH 7.0) - Total concentration of each essential oil in nanoemulsion was 25 mg/g.	[104]
Cosmetics and food	Thyme oil (thymol and carvacrol)	*S. aureus* and *P. digitatum*	Saponin (solvent and emulsifier)	[105]
Stored food items	Origanum majorana essential oil (terpinen-4-ol)	Fungi, aflatoxin B1 (AFB1) produced by *A. flavus*	Chitosan (deacetylation degree >85%), dichloromethane (DCM), dimethyl sulfoxide (DMSO), tripolyphosphate (TPP), anhydrous acetic acid, Tween 80, Tween 20, methanol, perchloric acid, sodium carbonate, chloroform,	[106]
Stored food mite	Ocimum basilicum (methyl eugenol, α-cubebene, linalool), Achillea fragrantissima (cis-thujone, 3,3,6-trimethyl-1,5-heptadien-4-one, 2,5-dimethyl-3-vinyl-4-hexen-2-ol, and trans-thujone), Achillea santolina (fragranyl acetate (26.1%), 1,6-dimethyl-1,5 cyclooctadiene (12.6%), 1,8 cineole (11.8%), and cis-thujone)	*T. putrescentiae* (Schrank)	Surfactant (Tween 80) as a non-ionic surfactant and deionized water at a ratio of 1:2:7	[107]
Minas Padrão cheese	Origanum vulgare essential oil (constituents not reported)	*Cladosporium* sp., *Fusarium* sp., and *Penicillium* sp. Genera	Sunflower oil, surfactants, deionized water, and oregano essential oil in two formulations: - Cremophor RH 40 (9.75%) and Brij 30 (3.25%) - Cremophor RH 40 (12%) and Span 80 (8%)	[108]
Edible coatings for fruits and vegetables (tomatoes)	Citrus sinensis essential oil (not reported)	*S. typhi* and *L*. *monocytogenes*	Sodium alginate 10 g L^−1^, Tween 80 2% (*w*/*v*)	[109]
Mayonnaise	Thymus daenensis L. essential oil (thymol and linalool)	*S. Typhimurium, E. coli*, and *L. monocytogenes*	Essentialoil:Tween 80, ratio 1:1, 15 min sonication	[110]
Aqueous food systems, beverages, and dairy	Black cumin essential oil (thymoquinone, longifolene, p-cymene, β-pinene, borneol, α-pinene, and α-thujene)	Two Gram-positive bacterial (GPB) strains *(B. cereus* and *L. monocytogenes)*	Pure CO or FSO or mixture of blackcurrant with canola and flaxseed oil at different ratios (2:8, 4:6, 6:4, and 8:2, respectively), plus octenyl succinic anhydrite modified starch	[111]
Fruit juices	Cold-pressed sweet orange (Citrus sinensis) essential oil (monoterpene hydrocarbons, oxygenated monoterpenes, sesquiterpene hydrocarbons, aliphatic aldehydes, myrcene, α-Pinene, sabinene, β-pinene, δ-3-carene)	*E. coli*	3 mL of Tween 80 with 10 mL of Citrus sinensis essential oil	[112]
Edible coatings	Clove and lemongrass essential oils, citral and eugenol component	*E. coli* and *B. cinerea*	Tween 80 (surfactant), food-grade sodium alginate	[113]
Edible films for meat products	Cinnamon essential oil (terpene mixture and D-limonene)	Gram-negative (*E. coli, P. aeruginosa,* and *S. typhi*) and Gram-positive *(E. faecium, B. cereus* and *S. aureus)*	First homogenizing 2 wt% cinnamon oils with 98% and aqueous emulsifier solution (1% *w*/*v* Soya protein isolate and 0.05 wt% lecithin) in high-speed blender	[114]
Functional food during storage	Zingiber zerumbet essential oil (camphene, eucalyptol, cis-geraniol)	*A. flavus,* aflatoxin B1 (AFB1)	Tween-80	[115]
Edible packaging for dairy and fruits	Clove essential oil (eugenol)	*S. aureus, E. coli*	Tween 80 and pectin (film)	[116]

**Table 4 nanomaterials-12-01196-t004:** Applications of nanoliposomes in the food industry.

Application	Targeted Microorganisms	Encapsulated Preservative	References
Food contact surfaces	*S. aureus, L. monocytogenes*, *E. coli*, and *Salmonella* spp.	Carvacrol	[120]
Food contact surfaces	*S. aureus* or *S. enterica*	Thymol, carvacrol and thymol/carvacrol	[121]
Milk, yogurt, spices, juice, processed meat, mayonnaise, and tahina	*E. coli, Salmonella,* and *Candida*	clove oil, black seed oil, thyme oil, garlic oil, rosemary oil, and green tea, tetracycline	[122]
Edible films	*S. aureus*	Nettle (Urtica dioica L.) extract	[123]
Functional foods, e.g., dairy products and beverages	*S. aureus, L. monocytogenes,* and *E. faecalis*	Nisin	[124]
Minas fresca cheese	*L. monocytogenes*	Nisin	[125]
Milk, dairy industry	*L. monocytogenes, S. Enteritidis,* *E. coli,* and *S. aureus*	Nisin and garlic extract	[126]
Tofu	*S. aureus* and *E. coli*	Clove oil	[127]
Milk containers	*S. aureus* biofilms	Salvia oil	[128]
Not reported	*(S. aureus, E. coli, S. Typhimurium,* and *L. monocytogenes*	Eugenol	[129]

**Table 5 nanomaterials-12-01196-t005:** Applications of SLNs and NLCs in the food and cosmetics industries.

Lipid Nanostructure	Incorporated Substance	Application	References
SLNs	Vitamin E	Cosmetic	[138]
SLNs	Quercetin	Food	[139]
NLCs	Mediterranean essential oils	Cosmetic	[140]
NLCs	Phenylethyl resorcinol	Cosmetic	[141]
NLCs	Retinol	Cosmetic	[142]
SLNs	Tretinoin	Cosmetic	[143]
SLNs	Coenzyme Q10	Cosmetic	[144]
SLNs and NLCs	Lycopene	Food	[145]
SLNs	Adenosine	Cosmetic	[146]
SLNs and NLCs	Resveratrol	Food and Cosmetic	[147,148]
SLNs	Citral	Food	[149]
SLNs	Mosquito repellent essential oils	Cosmetic	[150]
SLNs and NLCs	Alpha-lipoic acid	Cosmetic	[151]
SLNs	Carvacrol	Food	[152]
SLNs and NLCs	Butyl 4-hydroxybenzoate	Cosmetic	[153]

**Table 6 nanomaterials-12-01196-t006:** Applications of SLNs and NLCs in cosmetics and food preservation.

Lipid Nanostructure	Loaded Preservative	Emulsifiers and Surfactants	Target Microorganism	Result	References
SLNs	Parabens	Glyceryl distearate	*C. albicans*	Sustained release of parabens	[153]
NLCs	Parabens	Glyceryl distearate and almond oil	*C. albicans*	Sustained release of parabens	[153]
SLNs	Nisin	Glycerol monostearate 40–55% and Poloxamer 188	*L. monocytogenes* and *L. plantarum*	Prolonged release of nisin	[161]
SLNs	Carvacrol	Propylene glycol monopalmitate (PGMP) and Glyceryl monostearate (GMS) mixtures	*E. coli* and *S. aureus*	The 2:1 and 1:1 mass ratios of PGMP:GMS were feasible to prepare stable SLNs with enhanced antimicrobial activities.	[152]

## Data Availability

No new data were created or analyzed in this study.

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
