# Peer review of "Applications and Prospects of Nanotechnology in Food and Cosmetics Preservation"

_nanomaterials, 2022, doi:10.3390/nano12071196_

Round 1

Reviewer 1 Report

1.Please add previous studies about the toxicity of these nanomaterial's.
2.It will be great if authors add section about the in vitro effect of these nanomaterial's on cell sub-organelles.
3.Please add your future perspective and suggestion about the possible role of antioxidant therapy.
4.Please add more pictures about  the role of the Nanocellulose-based nanogels on cellular pathways.
5.It is suggested to use these papers for discussion part and bold the novelty of your study :

Eftekhari, Aziz, et al. "The promising future of nano-antioxidant therapy against environmental pollutants induced-toxicities." Biomedicine & Pharmacotherapy 103 (2018): 1018-1027.
-Ahmadian, Elham, et al. "The potential applications of hyaluronic acid hydrogels in biomedicine." Drug research 70.01 (2020): 6-11.

Author Response

We would like to thank the reviewer for the valuable comments.

Please see the answers on each point:

1.Please add previous studies about the toxicity of these nanomaterial's.

Line 434 Page 10 was added.

According to EFSA, approximately the 20% of E174 is used from confectionery pearls. This is the reason why Narciso et al. contacted a study to investigate the accumulation of silver nanoparticles in gastrointestinal system and its possible toxicological effect. According to this study, silver nanoparticles didn’t cause tissue damage nor genotoxicity even though was accumulated on the emptied duodenum due to the fact that they never passed into the cell nucleus […]

Narciso, L., Coppola, L., Lori, G., Andreoli, C., Zjino, A., Bocca, B., … Tassinari, R. Genotoxicity, biodistribution and toxic effects of silver nanoparticles after in vivo acute oral administration.NanoImpact, Volume 18,2020,100221

doi:10.1016/j.impact.2020.10022

2.It will be great if authors add section about the in vitro effect of these nanomaterial's on cell sub-organelles.

Page 2, line 87 was added.

Toxicity of nanomaterials is a hot topic which influences and affects both research studies and regulatory status. Nanoparticles can enter the human body most often through the respiratory or gastrointestinal tract, skin or through systemic administration depending on the route of exposure.

Several studies prove the accumulation of nanoparticles on different human organs namely liver, lungs, spleen and others. On a cellular level, nanoparticles when in contact with cells, are normally ingested by endocytosis, and are captured inside endosomes or lysosomes without entering cytoplasm.

In order to take chemical information and to support in vitro techniques that are used to study the distribution of nanoparticles into cells, SR-X-ray absorption near edge structure (XANES)can be applied. For example, this technique showed that silver nanoparticles were gradually transformed into ionic silver inside cultured macrophages.

 In vitro studies have conducted to show the effect of nanoparticles accumulation on cell cytoskeleton. Organization of cytoskeletal actin or F-actin orientation after the exposure of human fibroblasts on gold nanoparticles, have been defined by flow cytometry. 4 

Αt this point it is important to emphasize that the greatest destructive effect of nanoparticles when entering the cell is the free radicals production which is found to be increased at 350% when exposed on silver nanoparticles for 48hours. More specifically, that effect causes mitochondrial membrane disruption and cell viability reduction as a consequence. In a recent study depolarization phenomenon on mitochondrial membrane due to silver nanoparticles reaction, was visualized by flow cytometry and for cellular viability reduction scientists applied the Viability assay WST-8.5

3.Please add your future perspective and suggestion about the possible role of antioxidant therapy.

Page 25, Line 881 was added.

Finally the possible multifunctionality of the nano-preservative system is a factor that should be taken into consideration when designing contemporary foods or cosmetics. Ideal nano-ingredients could be designed to bear both efficacy for the bulk, acting as preservatives, and for the user, acting as bioactive ingredients. Indeed, antioxidants molecules, formulated into rationally designed nano-forms could both for example scavenge free radicals that deteriorate the quality of the formulation and act as nano-antioxidants that offer protection to the user from environmental aggressors.

4.Please add more pictures about  the role of the Nanocellulose-based nanogels on cellular pathways.

We appreciate the comment of the reviewer; however, the scope of our review is not to present the role of nanocellulose or other nanomaterials on cellular pathways but to present their applications and prospects for food and cosmetics preservation. As thus, we prefer to not add such specialized pictures.

5.It is suggested to use these papers for discussion part and bold the novelty of your study:

Eftekhari, Aziz, et al. "The promising future of nano-antioxidant therapy against environmental pollutants induced-toxicities." Biomedicine & Pharmacotherapy 103 (2018): 1018-1027. Έγινε στο κομμάτι του antioxidant therapy
-Ahmadian, Elham, et al. "The potential applications of hyaluronic acid hydrogels in biomedicine." Drug research 70.01 (2020): 6-11.

The following papers were added as reference.

Eftekhari, Aziz, et al. "The promising future of nano-antioxidant therapy against environmental pollutants induced-toxicities." Biomedicine & Pharmacotherapy 103 (2018): 1018-1027. Page 25, Line 881, Reference 197

-Ahmadian, Elham, et al. "The potential applications of hyaluronic acid hydrogels in biomedicine." Drug research 70.01 (2020): 6-11. Page 21, Line 641

Reviewer 2 Report

Dear Authors,

 The review article is interesting and very elaborative. I accept with minor corrections. Finally, cut out repetitions, run a spell-checker, and have it revised.

P1, line10: Cross-check the email address provided. Is that the right one?

P13 : Remove the highlight wt% , also P14: reference 106

Abbreviations of all the content should be provided. Spacing between the table and table legends need to be consistent.

All the table legends need to be in the same format. Example, table 2 legend and table 3 legend format is different. 

Author Response

We would like to thank the reviewer for the valuable comments.

Please see the answers on each point:

The review article is interesting and very elaborative. I accept with minor corrections. Finally, cut out repetitions, run a spell-checker, and have it revised.

The manuscript was checked for spelling and errors were corrected. Also repetitions were cut out.

Cross-check the email address provided. Is that the right one?

P1, line10: Corrected stagiaouris@aegean.gr

P13 : Remove the highlight wt% -

Corrected

also P14: reference 106 -done

Abbreviations of all the content should be provided. Spacing between the table and table legends need to be consistent.

Corrected

All the table legends need to be in the same format. Example, table 2 legend and table 3 legend format is different. 

Corrected

Reviewer 3 Report

The manuscript “Applications and prospects of nanotechnology in food and cosmetics preservation” is a review on the use of nanotechnology in food and cosmetics for preservation purposes and to promote their safer distribution. This is an accurate and good work related to hot topics. However, food and cosmetics could be divided in two completely separated sections of the work. Moreover, some revisions are required, as follows:

- Abstract. Add the main conclusions of this review.

- Among nano-vesicles, niosomes can be described. For this purpose, see for instance these recent works: 10.1016/j.jcou.2021.101669; 10.1080/1061186X.2022.2032094; etc…

- References are not in the journal style.

- Correct typos.

Author Response

Dear reviewer, thank you very much for your comments, your input is very important for us.

Please find our elaboration on your comments:

-The manuscript “Applications and prospects of nanotechnology in food and cosmetics preservation” is a review on the use of nanotechnology in food and cosmetics for preservation purposes and to promote their safer distribution. This is an accurate and good work related to hot topics. However, food and cosmetics could be divided in two completely separated sections of the work. 

Your suggestion was our primary approach. However, during our research we concluded that by separating the two sections, there would be a disproportion between them as most of the publications concern the category of food. Apart from that, in several publications, mainly on packaging, the author referred to common food and cosmetic applications.

Moreover, some revisions are required, as follows:

- Abstract. Add the main conclusions of this review.

Page 1, line 26 was added:

Conclusion: Nanomaterial’s science has already made a significant contribution to the food and cosmetics’ preservation and rapid developments in the last years reinforce the belief that in the future much of the preservation strategies to be pursued by the two industries will be based on nanoparticles and their nanocomposites.

- Among nano-vesicles, niosomes can be described. For this purpose, see for instance these recent works: 10.1016/j.jcou.2021.101669; 10.1080/1061186X.2022.2032094; etc…

Section 2.6.3. line 590, page 18 references 133-140 were added.

Niosomes can be mentioned as an advanced version of liposomes. They are delivery vehicles with closed bilayer structure, composed of non-ionic surfactants. They are recognized as safer and cheaper than liposomes, more stable with a longer self-life [1,2]. Nanoliposomes are prone to oxidation due to sensitive phospholipids [3]. On the other hand, niosomes face leakage deficiencies, a phenomenon that can be corrected by the addition of cholesterol [4]. Applications of niosomes can be found in both cosmetic and pharmaceutical industries taking into advantage their property to enhance bioavailability of bioactive compounds [5].  They are usually applied for the protection of sensitive compounds such us vitamin A, Ε or resveratrol (5,6,7). Interestingly, niosomes provide a protective delivery system for antibiotics making them remarkably effective in orthopedic, orthodontic, ophthalmological and other treatments [8] Applications in food industry are very few and their use in a preservation system has not studied yet.

  1. Muzzalupo R, Mazzotta EJEoodd. Do niosomes have a place in the field of drug delivery? Expert Opin Drug Deliv. 2019;16(11):1145–1147
  2. Bekraki AI. Liposomes-and niosomes-based drug delivery systems for tuberculosis treatment. Nanotechnology based approaches for tuberculosis treatment. 2020. India: Elsevier. p. 107–122. DOI: 10.1016/B978-0-12-819811-7.00007-2
  3. Kazi, K. M., Mandal, A. S., Biswas, N., Guha, A., Chatterjee, S., Behera, M., & Kuotsu, K. (2010). Niosome: A future of targeted drug delivery systems. Journal of advanced pharmaceutical technology & research, 1(4), 374–380. https://doi.org/10.4103/0110-5558.76435
  4. Rogerson A. Adriamycin-loaded niosomes–drug entrapment, stability and release. J Microencap. 1987;4:321–8
  5. Matos, M., Pando, D., & Gutiérrez, G. (2019). Nanoencapsulation of food ingredients by niosomes. Lipid-Based Nanostructures for Food Encapsulation Purposes, 447–481. doi:10.1016/b978-0-12-815673-5.00011-8 
  6. Machado ND, Gutiérrez G, Matos M, Fernández MA. Preservation of the Antioxidant Capacity of Resveratrol via Encapsulation in Niosomes. Foods. 2021 Apr 30;10(5):988. doi: 10.3390/foods10050988. PMID: 33946473; PMCID: PMC8147147.
  7. Basiri, L., Rajabzadeh, G., & Bostan, A. (2017). α-Tocopherol-loaded niosome prepared by heating method and its release behavior. Food Chemistry, 221, 620–628. doi:10.1016/j.foodchem.2016.11.12
  8. Mehrarya M, Gharehchelou B, Haghighi Poodeh S, Jamshidifar E, Karimifard S, Farasati Far B, Akbarzadeh I, Seifalian A. Niosomal formulation for antibacterial applications. J Drug Target. 2022 Jan 31:1-18. doi: 10.1080/1061186X.2022.2032094. Epub ahead of print. PMID: 35060818.

- References are not in the journal style.

Corrected

-Correct typos

Corrected